# NKCC1 Deficiency in Forming Hippocampal Circuits Triggers Neurodevelopmental Disorder: Role of BDNF-TrkB Signalling

**DOI:** 10.3390/brainsci12040502

**Published:** 2022-04-15

**Authors:** Jacek Szymanski, Liliana Minichiello

**Affiliations:** Department of Pharmacology, University of Oxford, Oxford OX1 3QT, UK; jacek.szymanski@pharm.ox.ac.uk

**Keywords:** GABA, GABA shift, NKCC1, BDNF, NTRK2/TrkB, neurodevelopmental disorders

## Abstract

The time-sensitive GABA shift from excitatory to inhibitory is critical in early neural circuits development and depends upon developmentally regulated expression of cation-chloride cotransporters NKCC1 and KCC2. NKCC1, encoded by the *SLC12A2* gene, regulates neuronal Cl^−^ homeostasis by chloride import working opposite KCC2. The high NKCC1/KCC2 expression ratio decreases in early neural development contributing to GABA shift. Human SLC12A2 loss-of-function mutations were recently associated with a multisystem disorder affecting neural development. However, the multisystem phenotype of rodent *Nkcc1* knockout models makes neurodevelopment challenging to study. Brain-Derived Neurotrophic Factor (BDNF)-NTRK2/TrkB signalling controls KCC2 expression during neural development, but its impact on NKCC1 is still controversial. Here, we discuss recent evidence supporting BDNF-TrkB signalling controlling *Nkcc1* expression and the GABA shift during hippocampal circuit formation. Namely, specific deletion of *Ntrk2*/*Trkb* from immature mouse hippocampal dentate granule cells (DGCs) affects their integration and maturation in the hippocampal circuitry and reduces *Nkcc1* expression in their target region, the CA3 principal cells, leading to premature GABA shift, ultimately influencing the establishment of functional hippocampal circuitry and animal behaviour in adulthood. Thus, immature DGCs emerge as a potential therapeutic target as GABAergic transmission is vital for specific neural progenitors generating dentate neurogenesis in early development and the mature brain.

## 1. Introduction

The Na-K-2Cl cotransporter 1 (NKCC1) belongs to the Solute Carrier Family 12 (SLC12) cation-chloride cotransporters (CCCs) and is encoded by the *SLC12A2* gene. The SLC12 family cotransporters mediate the electroneutral symport of chloride, potassium, and/or sodium across the plasma membrane. The family comprises nine members: *SLC12A1* (NKCC2), *SLC12A2* (NKCC1), *SLC12A3* (NCC), *SLC12A4* (KCC1), *SLC12A5* (KCC2), *SLC12A6* (KCC3), *SLC12A7* (KCC4) and two orphan members *SLC12A8* (CCC9) and *SLC12A9* (CIP), whose functional role and molecular structure is not well characterized [1]. The *SLC12A1-7* family plays a key role in maintaining Cl^−^, K^+^, and Na^+^ homeostasis by transepithelial ion absorption and secretion. They are critical for several physiological processes such as hearing, regulation of blood pressure, cell volume, and notably neuronal excitability [2,3,4,5,6,7]. Sodium-coupled CCCs (NCC, NKCC1, and NKCC2) are chloride importers activated by phosphorylation that raise the intracellular chloride concentration ([Cl^−^]_i_) through an inward sodium gradient. On the other hand, activated by dephosphorylation sodium independent CCCs (KCC1-4) extrude chloride, lowering the [Cl^−^]_i_ through an outward potassium gradient [7,8,9]. NKCC1 is expressed in both epithelial and non-epithelial cells and exhibits high conservation among species with more than 32 orthologues described and 94% sequence identity between mice and humans [10,11]. The other sodium-coupled CCCs (NCC and NKCC2) are mostly expressed in the kidneys where they regulate salt reabsorption [12], whereas KCC2 is neuron-specific and expressed in the mature central nervous system (CNS) [13].

In the mammalian CNS γ-aminobutyric acid (GABA) exerts both excitatory and inhibitory action on target cells through depolarization and hyperpolarization, respectively. GABA binds the ionotropic GABA_A_ receptor which as a ligand-gated ion channel allows for chloride influx or efflux through the plasma membrane, depending on the [Cl^−^]_i_. NKCC1 and KCC2 regulate Cl^−^ homeostasis in neurons of the CNS by importing and extruding chloride, respectively. In early neurodevelopment, low expression of KCC2 leads to a high NKCC1/KCC2 expression ratio and accumulation of Cl^−^ in the immature neurons. Under these conditions, GABA activation causes Cl^−^ efflux and membrane depolarization. With maturation, an ontogenetic shift in GABAergic signalling effect, from depolarizing and excitatory to hyperpolarizing and inhibitory, is observed. This is due to a higher expression of KCC2 with or without downregulation of NKCC1 which leads to a lower NKCC1/KCC2 expression ratio and a lower [Cl^−^]_i_, therefore, GABA activation results in Cl^−^ influx and membrane hyperpolarization [6,14,15,16]. In rodents, the upregulation of KCC2 mRNA and protein levels happens during the second postnatal week [6,17], while in humans the same occurs around full-term birth [6,18]. Early postnatal depolarizing GABA acts on neural precursors and immature neurons to drive proliferation, migration, and differentiation. Meanwhile, the hyperpolarizing GABA is required later to regulate activity and optimize information processing [19].

Among the different transcriptional regulators of CCCs, Brain-Derived Neurotrophic Factor (BDNF) plays an important role. *Bdnf* mRNA expression increases in the developing brain at different times, depending on the region. For example, it is low in the hippocampus at embryonic day 17 (E17) and gradually increases to reach high levels in the adult. In the cerebellum, it starts to increase at postnatal day 11 (P11), while in the spinal cord, it begins to rise at E12, peaking at P0 to further decrease in the adult brain [20].

BDNF belongs to a family of neurotrophins which are secreted molecules. It controls a variety of functions both in the peripheral and in the CNS, such as neuronal survival, cell fate decision, dendritic growth, as well as neuronal function by, for example, controlling the expression of ion channels [21]. In the adult brain, BDNF binding to its high-affinity receptor, NTRK2/TrkB, activates different signal transduction pathways leading to the regulation of synaptic plasticity and learning [21,22,23]. BDNF-TrkB signalling is well-known as one of the most critical regulators of glutamatergic and GABAergic synapse development [24,25]. BDNF and its precursor proBDNF regulate GABAergic neurotransmission by controlling GABA_A_ receptors’ cell membrane expression that is dependent on phosphorylation. Dephosphorylation of ß3 subunits of GABA_A_ receptors leads to their trafficking to endosomal compartments through interaction with the assembly polypeptide 2 (AP2) [26]. BDNF-TrkB signalling activates the protein kinase C (PKC) and phosphoinositide-3 (PI-3) kinase pathways, which inhibit GABA_A_ receptors’ dephosphorylation, thus preventing their internalization and increasing their cell surface expression [27]. Interestingly, Riffault et al. [28] have shown that proBDNF through its interaction with the p75 neurotrophin receptor (p75^NTR^) increases ß3 subunit GABA_A_ receptors internalization, thus decreasing their cell surface expression through the RhoA-Rock-PTEN pathway in cultured rat hippocampal neurons. On a transcriptional level, BDNF activates the cAMP-response element (CRE)-binding protein (CREB) through the ERK-MAP cascade that in turn regulates the transcription of GABA_A_ receptor subunits [29]. Meanwhile, the JAK-STAT pathway induced by proBDNF-p75^NTR^ leads to the downregulation of the ß3 subunit of the GABA_A_ receptor [28]. BDNF-TrkB signalling also modulates KCC2 expression in hippocampal CA1 pyramidal neurons through the Shc and PLCγ cascades, thus impacting [Cl^−^]_i_ and GABAergic transmission [30].

While the effect of BDNF-TrkB signalling controlling the expression of KCC2 during neuronal development and consequently the GABAergic shift has been widely described [31,32,33], the impact of this signalling on NKCC1 modulation is still controversial. In this review, we will focus our attention on recent evidence supporting such a role for BDNF-TrkB signalling during hippocampal circuit formation. We will also discuss the possible therapeutic potential of re-establishing the excitatory/inhibitory (E/I) homeostasis in particular cell types to revert or ameliorate neurodevelopmental disorders.

## 2. Transcriptional Regulation of *Nkcc1*

Data on NKCC1 expression in the CNS is often contradictory, with reports of NKCC1 downregulation, upregulation, as well as stable expression, independently of the brain region studied [11]. However, there have been some compelling studies describing the accumulation of Cl^−^ by NKCC1 during the development of central neurons. One study using in vivo two-photon imaging obtained the combined measurement of Cl^−^ and pH of single mouse cortical pyramidal neurons from early postnatal development to maturation, allowing to gather direct evidence involving the NKCC1 transporter maintaining high [Cl^−^]_i_ in immature neurons and the developmental somatic decrease of [Cl^−^]_i_ from P4–P5 to P8–P10, with a further decrease in neuronal maturation at P18–P51. These data were further validated using a blocker of the NKCC1 transporter (bumetanide) [34]. Another study has provided evidence for the excitatory action of interneurons on neonatal hippocampal neurons in vivo [35]. Finally, it has also been shown that NKCC1 is responsible for high [Cl^−^]_i_ found in neuronal progenitors, including the dentate gyrus [36].

Previous studies have also shown that a single high dose, as opposed to multiple injections of BDNF, induced downregulation of NKCC1 in the rat hippocampus of pilocarpine-induced status epilepticus [37]. However, in a more recent study, Badurek et al. [38] show that specific deletion of *Ntrk2*/*Trkb*, the BDNF high-affinity receptor coding gene, from immature mouse hippocampal dentate granule cells (DGCs) leads to reduced expression of *Nkcc1* in their target region, the CA3 principal cells of the mouse hippocampus (Figure 1A–E), and to lower [Cl^−^]_i_, without changes to KCC2 expression; which drives a premature GABA shift from depolarizing to hyperpolarizing at mossy fibers (MF)-CA3 synapses (Figure 1F–I) [38]. *Nkcc1* reduced expression was confirmed by single-molecule fluorescence in situ hybridization (smFISH), a powerful technique allowing to study gene expression with higher precision (Figure 1A–E).

Moreover, the premature shift of GABA to inhibitory transmission impaired morphological maturation and synaptic connectivity of the CA3 target neurons, ultimately affecting the formation of proper hippocampal circuitry and adult synaptic plasticity and cognition [38]. These data provide in vivo evidence for BDNF-TrkB signalling transcriptional regulation of NKCC1 transporter in immature neurons. Although the exact mechanism by which presynaptic TrkB signalling in immature dentate granule neurons regulates postsynaptic *Nkcc1* transcription in CA3 pyramidal neurons requires further investigation, the data establish the genetic importance of TrkB signalling in immature DGCs, driving the sequential development of intrinsic hippocampal circuits by modulating early GABA signalling through the expression of *Nkcc1*.

Among other regulators of chloride homeostasis in the CNS, oxytocin modulates the function of NKCC1 during development [15,39]. In the perinatal period, the fetal hippocampus expresses high levels of the oxytocin receptor, which is reached by the highly expressed maternal oxytocin at delivery. Oxytocin causes a transient GABA shift to inhibitory before delivery, associated with decreased NKCC1 activity in CA3 neurons of the hippocampus of rodent fetal brains [39].

## 3. *SLC12A2* Mutations Reported in Humans and Their Involvement in Neurodevelopmental Disorders

GABAergic circuit dysfunction is a common feature of many neurodevelopmental disorders (NDs) [40,41]. The dysregulation of the GABA system in NDs through upregulation of NKCC1 and/or downregulation of KCC2 is well described in many diseases, both in rodents and humans [14,42,43,44,45,46]. However, little is known of the effects of NKCC1 downregulation during the depolarizing stage occurring in immature neurons. Recently, Macnamara et al. [47] described a case of a young 5-year-old child with a 22 kb deletion affecting exons 2–7 of *SLC12A2*, rendering the boy with complete absence of NKCC1 expression (Table 1). Since NKCC1 is broadly expressed throughout the body, the patient was severely affected with respiratory weakness, gastrointestinal issues, pancreatic exocrine dysfunction, growth disturbance, hearing loss, and notably intellectual disability. The boy inherited two copies of chromosome 5 from his father through uniparental disomy, suggesting an autosomal recessive condition that the authors named Kilquist syndrome. The second case of a 9-year-old girl [48] with a compound heterozygous mutation in *SLC12A2*, was described a year later (Table 1). The patient carried a one base substitution (c.2006-1G>A) which caused exon skipping and open reading frame disruption by affecting the splice acceptor site of exon 13, and a one-base deletion (c.1431delT) causing a premature termination codon by a frameshift mutation in exon 8. She had similar disease manifestations as the 5-year-old boy with severe intellectual disability and dysmorphic facial features, as well as respiratory weakness and gastrointestinal issues. She also had an older sibling, with the same ailments who carried the same mutations, but died at 2 months of age. Their parents were both carriers of one mutation suggesting the autosomal recessive Kilquist syndrome. Cystic fibrosis and metabolic diseases were tested for and excluded in all three patients. Another biallelic loss-of-function variant in *SLC12A2* was described in a girl with a c.940C>T, p.Q314* variant from her father and a splice donor variant c.1536+4_1536+7del, p.? from her mother [49] (Table 1). Her phenotype matched the Kilquist syndrome with a multisystem disorder of the respiratory system, gastrointestinal problems, and bilateral sensorineural hearing loss. Additional four cases of a homozygous splice variant c.2617-2A>G, p.? were described in two separate consanguineous families [50]. They also had a multisystem disorder with neurodevelopmental delay, microcephaly, subtle dysmorphic features, respiratory difficulties, and hearing loss in some. No gastrointestinal problems were reported. These cases suggest that the absence of NKCC1 expression is not immediately lethal but gravely burdensome and plays a role in affecting neurodevelopment in humans.

McNeill et al. [51] have described a link between de novo mutations in *SLC12A2* and NDs in six individuals aged 12 months to 21 years. They all had mild to severe intellectual disability or developmental delay. Some were diagnosed with autism and spastic paraparesis and some have had feeding difficulties. All carried single nucleotide variants (SNVs): g.127450305C>T, p.A327V (exon 4); g.127469897G>A, p.R410Q (exon 6); g.127503511G >A, p.W892* (exon 18); g.12746683A>T, p.N376I (exon 5); g.127420201dup, p.H186fs16 (exon 1); g.127466845G>C, p.A379L (exon 5) (Table 1). The pleiotropic effects and phenotypic manifestations were not as severe in these patients as in the ones with a complete absence of NKCC1 expression. In addition, the patients with missense mutations were less severely affected than those with truncating mutations. It is important to note that McNeill et al. [51] have also identified three patients with SNVs in exon 21 of *SLC12A2* who did not show signs of NDs but had bilateral sensorineural hearing impairment or loss. This is most likely the consequence of different spatial expressions of the two best-characterized splice variants of NKCC1. NKCC1a is a full-length variant and NKCC1b lacks exon 21. NKCC1b/NKCC1a ratio is much higher in the human brain [52], while NKCC1a but not NKCC1b is expressed in the mouse cochlea. Mutations in this variant lead to deafness in mice [53]. These results are supported by a study by Mutai et al. [54] who have identified 3 heterozygous mutations in exon 21 of *SLC12A2*, all associated with hearing loss. Two of those were missense mutations (p.D981Y and p.P988T) in exon 21 and one was a variant at the 3′-splice site of the exon 21 (corresponding to p.977_992del). The splice site variant leads to skipping of the exon 21 in the transcript. An in vitro functional study showed a reduced Cl^−^ influx in all three *SLC12A2* mutants and decreased level of mRNA for the 3′-splice site variant [54]. A SNV c.2589G>C, p.L863F was also associated with autism–epilepsy phenotype with macrocephaly with a severe intellectual disability through an in silico analysis of next-generation sequencing data [55]. Curiously, a SNV c.596A>G, p.Y119C in *SLC12A2* has been linked to schizophrenia as a gain-of-function missense mutation [56]. Functional experiments of c.596A>G, p.Y119C, with *Xenopus* oocytes and cRNA constructs, suggest an increase in NKCC1 activity and higher Cl^−^ uptake [56].

Another case of a de novo 11bp deletion in exon 22 resulting in a frameshift mutation (c.3076_3086delGTCTGGTGGCT, p.V1026Ffs*2) was described [57] in a 13-year-old female patient who also presented a multisystem disorder with respiratory weakness, several endocrine abnormalities, multiorgan failure but no ND. To recapitulate, lack of NKCC1 expression leads to severe symptoms, not only neurodevelopmental but, among others, gastrointestinal and respiratory. SNVs in *SLC12A2* are linked to NDs and variants specific to exon 21, present in NKCC1a, to hearing impairment or loss. A case can be made for both dominant and recessive *SLC12A2*-syndrome. McNeill et al. [49] suggest subtle neurocognitive or hearing phenotypes in heterozygous parents of patients with an autosomal recessive syndrome cannot be fully excluded. They also discuss the possible mislocalization or impairment of critical interactions in missense variants of *SLC12A2*, especially important since NKCC1 works as a dimer. They point out that loss-of-function variants of *SLC12A2* would be less likely to have a dominant-negative effect as it would be improbable to produce a mutant protein with a disrupted structure. However, the nonsense mutants may be cleared through nonsense-mediated mRNA decay leading to haploinsufficiency.

## 4. Chloride Cotransporter NKCC1 in Animal Models of Neurodevelopmental Disorders

Several *Nkcc1* knockout mouse models have been created [53,57,58,59]. These mutants are smaller than wild-type mice and are deaf with inner ear defects leading to imbalance. *Nkcc1* knockout causes debilitating effects on the vestibular system leading to spinning and head bobbing, making it challenging to study behaviour. NKCC1 also plays a role in fluid secretion in the intestines, the lungs, and the salivary gland [60,61,62]. Its absence in knockout mouse models leads to intestinal obstruction and a significant decrease in saliva secretion, while no respiratory disease has been described. Hyperkalemia was also observed as a renal phenotype of this model [63]. The multisystem disorder phenotype of *Nkcc1* knockout mice seems to reflect the ailments observed in humans; however, it’s difficult to study neurodevelopmental delays in these models. An additional hurdle of male sterility of knockout mice makes it impossible to cross homozygotes [64]. In the adult CNS of both mice and humans, NKCC1 is mainly expressed in the oligodendrocyte precursor cells with especially high expression in regions of the hippocampus and the choroid plexus [11]. NKCC1 mediates cerebrospinal fluid (CSF) clearance in the choroid plexus during early postnatal neurodevelopment [65]. This is important as abnormal accumulation of the CSF can cause neurodevelopmental problems. NKCC1 is also expressed in the sensory neurons of the dorsal root ganglia, and in the knockout mice, there is a shift to a hyperpolarizing GABA action in these cells, leading to lower pain sensitivity [66]. These are interesting data; however, it is challenging to study NDs using the full *Nkcc1* knockout rodent model due to the issues mentioned above. Therefore, more work is needed to elucidate the neurodevelopmental problems related to *SLC12A2* mutations causing a decrease in NKCC1 expression levels, as evident in humans. Of high interest is to understand their specific impact on the time-sensitive GABA shift in early neurodevelopment.

## 5. Impact of NKCC1 Expression on GABA Shift and Neural Circuits Development

Excitatory depolarizing GABA, inhibitory hyperpolarizing GABA, as well as the timing of the GABA shift are all essential for correct neuronal circuits development and further sensory input processing in adult life [19].

Both depolarizing GABA and glutamate mediate radial glia cells’ proliferation in the ventricular zone (VZ) and decrease the proliferation of intermediate progenitor cells in the subventricular zone (SVZ) as shown using organotypic slice cultures [67]. Depolarizing GABA also plays a role in immature neuron migration from the VZ and SVZ through the intermediate zone to the developing cortical plate (CP). It mediates neuronal migration through transiently expressed GABA_A_-ρ receptors, in the migrating immature neurons, to then act as a stop signal through GABA_A_ receptors at the CP [68]. Another role of depolarizing GABA in neuronal circuits development is its promotion of neurite growth and synapse formation and maturation [19,69]. Interestingly, GABA excitatory action not only promotes neuronal dendritic arborization and synapse formation in early brain development but also of newly generated neurons in the adult hippocampus [36].

Hippocampal giant depolarizing potentials (GDPs) are spontaneous neuronal oscillations, essential for several developmental processes mediated by GABAergic and glutamatergic depolarizations [70]. Inhibition of hippocampal KCC2 in perinatal rodents enhanced GABAergic depolarization and in turn hippocampal GDPs [71]. In a mouse model of idiopathic autism, GDPs are impaired in the CA3 region of the hippocampus, which is associated with decreased neuronal excitability [72]. Khalilov et al. [73] suggest GABAergic actions change from depolarizing to hyperpolarizing at the network level during GDPs in the hippocampus of neonatal rats, thus protecting it from epileptiform synchronization. Loss of depolarizing GABAergic action leads to epileptiform action in the cortex and hippocampus [73,74,75]. Thus, GABA action mediates GDPs through depolarization and limits them through its hyperpolarizing action at the network level. The same can be observed for sharp waves (SWs), an in vivo counterpart of GDPs, in the hippocampus [35,76].

The functions of depolarizing GABAergic action are therefore crucial for neurodevelopment. In addition, the timing of the GABA shift is extremely important, yet it seems dependent on the region of the brain, sex, and cell type. Hippocampal interneurons shift to inhibitory action at P7, while visual cortex interneurons shift at P3 [35]. In females, the GABA shift appears earlier than in males in hippocampal and midbrain slices [77,78,79] but later in the cerebellum [80]. As summarised by Peerboom and Wierenga [19], the change in [Cl^−^]_i_, an indicator of GABA shift, not only varies between neurons but also within single pyramidal neurons, as the GABAergic reversal potentials are more negative at the soma and less negative towards the dendrites and initial axon segment. Although variable, the timing of GABA shift is important for correct neuronal circuits formation and both delayed and advanced shift can be detrimental to neurodevelopment, as evidenced by epileptiform discharges depending on loss of depolarizing GABAergic action [73,74,75].

Wang and Kriegstein published two papers [81,82] where they advanced GABA shift through RNAi knockdown and inhibition of NKCC1 with bumetanide in the neocortex. Advancing hyperpolarizing GABAergic action in rodents in this manner had important consequences for glutamatergic synapses development. Blocking NKCC1 between E17 and P7, the GABA shift occurred as early as P0 leading to reduced α-amino-3-hydroxyl-5-methyl-4-isoxazole-propionate (AMPA) at synapses two to four weeks after birth. This caused a reduced excitatory synaptic transmission which persisted in adulthood. The premature GABA shift also led to improper dendritic development of cortical neurons and sensorimotor gating deficits. Interestingly, blocking NKCC1 between E15 and E19 or between P0 and P7 does not produce a persistent effect on the cortical glutamatergic synapses, highlighting the time sensitivity of the depolarizing, hyperpolarizing and GABA shift GABAergic actions. The reduction of glutamatergic synaptic transmission through advanced GABA shift caused by NKCC1 blockade can be rescued by the expression of voltage-independent N-methyl-D-aspartate (NMDA) receptors. This suggests GABA regulates cortical glutamatergic synapse formation via NMDA receptor activation. In a different experiment, GABA shift was advanced by premature expression of KCC2 in mouse organotypic slice culture and in utero in rats [83]. This led to a reduction of dendritic spine density in neurons of the CA1 region of the hippocampus but an increase in spine density in cortical neurons. Similarly, premature expression of KCC2 in *Xenopus* led to abnormal development of glutamatergic synapses [84]. Together, this shows the importance of GABAergic transmission in the maturation of glutamatergic synapses, whose dysfunction is associated with many NDs [85], via NMDA receptors in developing neuronal circuits across different species. Inhibition of NKCC1 also affects the development of inhibitory synapses in cultured neocortical neurons [86], yet overexpression of KCC2 increases GABAergic transmission [84,87], an effect not fully understood. An advanced GABA shift was also observed in the target region of DGCs, the CA3 principal cells of the mouse hippocampus, in concomitance with reduced *Nkcc1* expression driven by the specific deletion of *Trkb* from immature mouse hippocampal DGCs. No effect on KCC2 expression was observed [38]. Depletion of BDNF-TrkB signalling and reduced *Nkcc1* expression also led to disruption of GDPs. The immature DGCs exhibited delayed maturation and integration into the DGC layer and consequently affected the sequential maturation of intrinsic hippocampal circuits resulting in adult hippocampal dysfunctions. It is remarkable to point out that TrkB and its ligand, BDNF, are well-known as critical regulators of adult hippocampal functions independent of NKCC1 mediated effects [24,25,88,89]. In particular, the deletion of *Trkb* from forebrain principal cells at P20 [22], including dentate granule cells that have mainly reached and integrated into the granule cell layer, does not affect morphological and structural neuronal features but impairs hippocampal long-term potentiation (LTP) and learning primarily complex or stressful learning paradigms. Instead, the deletion of *Trkb* from immature DGCs early in postnatal development affects neurodevelopmental programs leading to adult neural dysfunction [38]. This shows that BDNF-TrkB signalling is critical in establishing E/I homeostasis at MF-CA3 synapses, as well as in the development of the hippocampal circuits playing a role in GABA shift. Interestingly, common neural progenitors (Hopx^+^ precursors) were found to generate dentate neurogenesis both early in development and in the adult mouse brain, as observed by single-cell lineage tracing and population fate-mapping [90]. GABAergic transmission is critical for newborn granule cells’ activation and integration, as well as innervation, together with the glutamatergic transmission, in the murine brain [36]. A premature GABA shift can lead to defects in granule cell synapse formation and dendritic development [36]. This opens a window of therapeutic opportunities for possible treatment of newly integrating immature granule cells, once the molecular mechanism of NKCC1 regulation by BDNF-TrkB signalling is better understood. This is especially true as NKCC1 expression in the human brain is significantly higher in VZ and SVZ than in regions of less active neurogenesis (CP, subplate), measured at 15–16 weeks of gestation [52].

Advanced GABA shift, induced by NKCC1 inhibition through bumetanide intraperitoneal injections of pregnant dams and postnatal pups, results in reduced locomotor activity and developmental delay in motor coordination and strength, as seen by less movement under observation, affected negative geotaxis, and bar holding [82]. Mice with earlier hyperpolarizing GABA also show lower anxiety than wild-type counterparts [82].

## 6. The Therapeutical Potential of Re-Establishing E/I Homeostasis of GABAergic Signalling

The major pharmacological focus in disrupted GABA shift due to increased NKCC1/KCC2 expression ratio is on re-establishing the E/I balance through NKCC1 inhibition. Bumetanide, an FDA-approved diuretic, and nonselective inhibitor of NKCC1 used to treat oedema, cardiac failure, pulmonary congestion, or hepatic and renal disease, such as nephrotic syndrome [14], has also been investigated as a possible treatment targeting NKCC1 upregulation in many neurological diseases [91,92]. Several symptoms related to autism spectrum disorders (ASDs) could be reversed by bumetanide use in mouse models of these disorders [93,94,95]. In addition, in Phase II clinical trials, bumetanide appears to ameliorate autism symptoms [91,92]. Bumetanide was found to be effective at improving symptoms of Down syndrome [44], schizophrenia [45,96], neonatal seizures [91], and epilepsy [97]. However, bumetanide does not easily penetrate the blood-brain barrier, which may make it less effective as a treatment for NDs [98]. The broad expression of NKCC1 throughout the body, bumetanide’s inhibitory action on NKCC2, and its diuretic function are also problematic as they may cause hypochloremia and hypokalemia, hyperuricemia, prerenal azotemia, and metabolic alkalosis [99]. Thus, bumetanide is a promising drug in GABA shift related NDs therapy [100] but generates health concerns for chronic treatment. Alternative NKCC1 inhibitors were also tested with some success [94,101]. For example, a newly discovered small molecule ARN23746, a selective inhibitor of NKCC1, was shown to improve Down syndrome and autism symptoms in murine models of these diseases and restore [Cl^−^]_i_ in in vitro murine Down syndrome neuronal cultures [94]. These results suggest a new potential drug candidate for neurological conditions based on impaired E/I homeostasis through NKCC1 inhibition.

Reports discussing NKCC1 mutations reducing its expression and leading to NDs are very recent [10,47,49,50,51,56], as is the established in vivo relationship between specific *Trkb* deletion from immature DGCs and decreased *Nkcc1* expression in a novel mouse model leading to neurodevelopmental disorder and affected cognitive impairment in adult [38]. Therefore, since there are only a few therapeutic choices to ameliorate this problem, it is critical to further investigate the relationship mentioned above between BDNF-TrkB signalling in immature neurons and NKCC1 regulation to identify novel therapeutic solutions.

## 7. Conclusions and Future Directions

NKCC1 and KCC2 expression dysregulation has emerged as an essential factor affecting GABAergic transmission shift from depolarizing to hyperpolarizing in early brain development. Early postnatal depolarizing GABA is crucial for proliferation, migration, and differentiation of neural precursors and immature neurons while hyperpolarizing GABA is needed for the optimization of sensory information processing. Early disruption of the [Cl^−^]_i_ homeostasis in the CNS development leads to an advanced or delayed GABA shift and to neurological and neurodevelopmental disorders [19,102]. This time-sensitive disruption alters early synchronous events, such as the GDPs in the developing hippocampus, altering proper neuronal circuit development. Mutations in the *SLA12A2* gene encoding NKCC1 have recently been described in patients suffering from a multisystem disorder affecting neurodevelopment, and intestinal and pulmonary obstruction among a plethora of ailments [10]. This is due to the broad NKCC1′s expression in many body tissues, including the CNS. Although *Nkcc1* knockout animal models make neurodevelopmental studies difficult due to a multisystem phenotype, many studies associate NKCC1 inhibition with advanced GABA shift and NDs [81,82,83,84]. Most rodent models focus on NKCC1 upregulation and KCC2 downregulation as drivers of GABAergic transmission-related disorders. Here, we described a novel mouse model with a selective deletion of *Trkb* from immature hippocampal DGCs which leads to a premature GABA shift from depolarizing to hyperpolarizing in the target of DGCs, CA3 principal cells of the mouse hippocampus [38]. This advanced shift occurred with decreased NKCC1 expression and lower [Cl^−^]_i_, without changes to KCC2 expression. As GABAergic transmission is critical for the correct development of specific neural progenitors which generate dentate neurogenesis in early development and mature CNS, an interesting therapeutic target emerges in the form of newly integrating immature granule cells. Therefore, the regulatory relationship between BDNF-TrkB activated pathway/s and NKCC1 requires further investigation.

## Figures and Tables

**Figure 1 brainsci-12-00502-f001:**
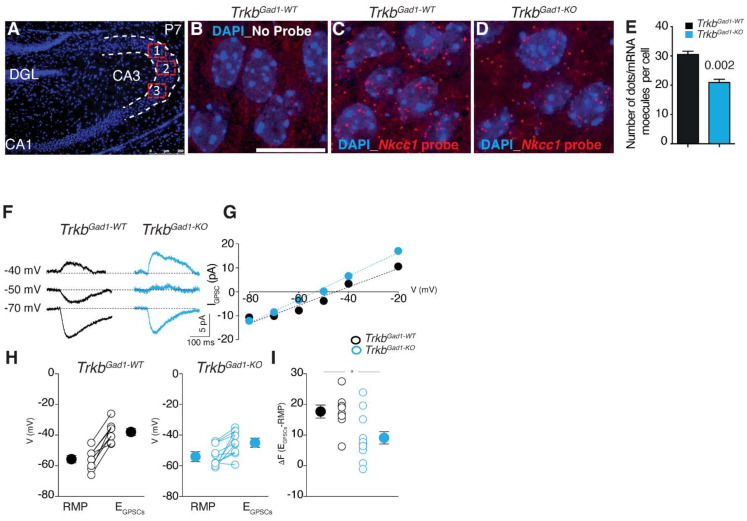
Reduced *Nkcc1* expression and altered direction of GABA at immature mossy fiber (MF)-CA3 synapses caused by *Trkb* deletion in immature DGCs. (**A**–**E**) Single-molecule fluorescence in situ hybridization (smFISH) is used to detect and count individual *Nkcc1* RNA molecules in single cells of the CA3 region at P7. (**A**) DAPI stained P7 hippocampal section highlighting three random fields imaged in the CA3 region for quantification of single-molecule RNA. (**B**–**D**) Representative images from the CA3 regions of control (*Trkb^Gad1-WT^)* and mutant (*Trkb^Gad1-KO^)* mice highlighting the single cells by DAPI nuclear staining. Single mRNA molecules (red spots) derived from the transcription of *Nkcc1* are detected with the Quasar570 fluorophore-labelled oligonucleotide probe library in single cells of the CA3 hippocampal region (**C**,**D**); no probe control (**B**). (**E**) Quantification of single mRNA molecules per cell *p* = 0.002; *n* = 3 P7 pups each genotype. DGL, dentate granule layer. Scale bars: 250 μm in (**A**) and 50 μm in (**B**–**D**). (**F**–**I**) The reduced driving force for GABA-mediated postsynaptic currents (GPSCs) at MF-CA3 synapses in *Trkb^Gad1-KO^* mice. (**F**) Representative traces of GPSCs were evoked at three different holding potentials in CA3 principal cells by MF stimulation (gramicidin-perforated patches) in controls and mutant mice. (**G**) Amplitudes of GPSCs (IGPSC) shown in (**F**) are plotted against holding potentials (V). (**H**) Individual RMPs and EGPSCs values in CA3 principal cells from control and *Trkb^Gad1-KO^*. Larger symbols on the left and right refer to mean G SEM values. (**I**) The plot of the driving force (DF) for GABA (DF = EGPSCs − RMP) in individual experiments. Larger symbols are mean G SEM values. * *p* = 0.03, Wilcoxon test. Adapted from Badurek et al., **2020**, iScience *23*, 101078 [38].

**Table 1 brainsci-12-00502-t001:** Human mutations decreasing *SLC12A2* expression.

Type of Mutation	Age	Effect of the Mutation on the Expression of *SLC12A2*	Phenotype	Reference
22 kb deletion affecting exons 2–7 of *SLC12A2*. chr5:127441491–127471419	5-year-old	complete absence of NKCC1 expression	multisystem disorder phenotype including intellectual disability	[47]
Biallelic loss-of-function variant in *SLC12A2* c.2006-1G>A and c.1431delT	2-month-old and 9-year-old	NKCC1 deficiency	multisystem disorder phenotype including intellectual disability	[48]
Biallelic loss-of-function variant in *SLC12A2* c.940C>T, p.Q314* and c.1536+4_1536+7del, p.?	1-year-old	NKCC1 deficiency	multisystem disorder phenotype including intellectual disability	[49]
de novo mutations, single nucleotide variants (SNVs): g.127450305C>T, p.A327V (exon 4); g.12746683A>T, p.N376I (exon 5); g.127420201dup, p.H186fs16 (exon 1); g.127469897G>A, p.R410Q (exon 6); g.127503511G>A, p.W892* (exon 18); g.127466845G>C, p.A379L (exon 5).	1-year-old, 3-year-old, 6-year-old, 9-year-old, 15-year-old, 21-year-old	NKCC1 deficiency	Mild to severe intellectual disability or developmental delay. Some were diagnosed with autism and spastic paraparesis and some have had feeding difficulties	[51]

## Data Availability

The data supporting the reported results can be found in the literature, appropriately cited here.

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
