# Peer review of "NKCC1 Deficiency in Forming Hippocampal Circuits Triggers Neurodevelopmental Disorder: Role of BDNF-TrkB Signalling"

_brainsci, 2022, doi:10.3390/brainsci12040502_

Round 1

Reviewer 1 Report

This is a comprehensive review of current knowledge on molecular and cellular mechanisms of the neurodevelopmental changes in GABAergic transmission. Furthermore, a potential role of these alterations in synaptic plasticity, learning and memory, as well as in some pathologies of CNS have been convincingly presented and discussed in detail. Specifically, the authors focuse on the role of NKCC1 and KCC2 cation-chloride cotransporters as key factors affecting GABA shift from depolarizing to hyperpolarizing in early brain development and emphasize an involvement of (BDNF)-NTRK2/TrkB signalling in controlling of KCC2 and NKCC1 gene expression. This topic is important because it suggests new avenues in treatment of some neurodevelopmental disorders.  Collectively, this review has been written in a professional way with a clear structure.  

Author Response

We thank the reviewer for their very positive comments.

Reviewer 2 Report

Comments to the paper: NKCC1 deficiency in forming hippocampal circuits triggers 2 neurodevelopmental disorder: role of BDNF-TrkB signaling

I would like the authors to improve the following issues:

  1. Introduction – there is no information on the potential substrates or inhibitors of NKCC1.
  2. Figure 1 – Do the authors have permission to incorporate Figure 1?
  3. Can the authors highlight the novelty of this paper?

Author Response

We thank the reviewer for their comments.

Response to Reviewer 3 Comments

I would like the authors to improve the following issues:

Point 1: Introduction – there is no information on the potential substrates or inhibitors of NKCC1.

Response 1: We believe that the primary substrates of NKCC1 being ions have already been mentioned in the introduction. As for the potential inhibitors, we have specifically paragraph 6 on page 9 where we talk about inhibitors of NKCC1. The main point is that there are not many inhibitors of NKCC1 and therefore we have focused our attention on the most widely described, bumetanide. However, we have now added some discussion on a new small-molecule inhibitor of NKCC1, ARN23746, that has shown high specificity in preclinical trials so far.  

Please, see the text highlighted on page 9, paragraph 6.

... Thus, bumetanide is a promising drug in GABA shift-related NDs therapy [100] but generates health concerns for chronic treatment. Alternative NKCC1 inhibitors were also tested with some success [94, 101]. In particular, a newly discovered small molecule ARN23746, a selective inhibitor of NKCC1, was shown to improve Down syndrome and autism symptoms in murine models of these diseases and restore [Cl-]i in in vitro murine Down syndrome neuronal cultures [94]. These results suggest a new potential drug candidate for neurological conditions based on impaired E/I homeostasis through NKCC1 inhibition. 

Point 2. Figure 1 – Do the authors have permission to incorporate Figure 1?

Response 2:  The figure comes from Open Access Journals (iScience), no
permission is needed.

Point 3. Can the authors highlight the novelty of this paper?

Response 3:  Please see the answer on Page 4

Although the exact mechanism by which presynaptic TrkB signalling in immature dentate granule neurons regulates postsynaptic Nkcc1 transcription in CA3 pyramidal neurons requires further investigation, the data establish the genetic importance of TrkB signalling in immature DGCs, driving the sequential development of intrinsic hippocampal circuits by modulating early GABA signalling through the expression of Nkcc1.

Reviewer 3 Report

Well-written review citing recent relevant literature. Very well focused and clear.

My only comment is to add discussion on potential non-NKCC1 mediated effects of BDNF/TrkB signalling on hippocampal development, maturation and function.

Author Response

We thank the reviewer for their comments.

Response to Reviewer 3 Comment

Response 1: We believe that adding discussion about the potential non-NKCC1 mediated effects of BDNF/TrkB signalling on hippocampal development, maturation and function is out of this review focus. However, we agree to emphasize the adult hippocampal function and to expand this important concept on pages 8-9 where we already introduced this concept.

Depletion of BDNF-TrkB signalling and reduced Nkcc1 expression also led to disruption of GDPs. The immature DGCs exhibited delayed maturation and integration into the DGC layer and consequently affected the sequential maturation of intrinsic hippocampal circuits resulting in adult hippocampal dysfunctions. It is remarkable to point out that TrkB and its ligand, BDNF, are well-known as critical regulators of adult hippocampal functions independent of NKCC1 mediated effects [24, 25, 88, 89]. In particular, the deletion of Trkb from forebrain principal cells at P20 [22], including dentate granule cells that have mainly reached and integrated into the granule cell layer, does not affect morphological and structural neuronal features but impairs hippocampal long-term potentiation (LTP) and learning primarily complex or stressful learning paradigms. Instead, the deletion of Trkb from immature DGCs early in postnatal development affects neurodevelopmental programs leading to adult neural dysfunction [38].

Reviewer 4 Report

This is an interesting review but it could have been expanded a bit more on the exact role of BDNF on GABA and Cl channels. It would be nice if authors could highlight some mechanism and pathways involved in this process. 

Author Response

We thank the reviewer for their comments.

Response to Reviewer 4 Comments

Response 1: introduction, page 2.

BDNF-TrkB signalling is well-known as one of the most critical regulators of glutamatergic and GABAergic synapse development [24, 25]. In particular, BDNF and its precursor proBDNF regulate GABAergic neurotransmission by controlling GABAA receptors’ cell membrane expression that is dependent on phosphorylation. Dephosphorylation of ß3 subunits of GABAA receptors leads to their trafficking to endosomal compartments through interaction with the assembly polypeptide 2 (AP2) [26]. BDNF-TrkB signalling activates the protein kinase C (PKC) and phosphoinositide-3 (PI-3) kinase pathways, which inhibit GABAA receptors’ dephosphorylation, thus preventing their internalization and increasing their cell surface expression [27]. Interestingly, Riffault et al. [28] have shown that proBDNF through its interaction with the p75 neurotrophin receptor (p75NTR) increases ß3 subunit GABAA receptors internalization, thus decreasing their cell surface expression through the RhoA-Rock-PTEN pathway in cultured rat hippocampal neurons. On a transcriptional level, BDNF activates the cAMP-response element (CRE)-binding protein (CREB) through the ERK-MAP cascade that in turn regulates the transcription of GABAA receptor subunits [29]. Whereas the JAK-STAT pathway induced by proBDNF-p75NTR leads to the downregulation of the ß3 subunit of the GABAA receptor [28]. BDNF-TrkB signalling also modulates KCC2 expression in hippocampal CA1 pyramidal neurons through the Shc and PLCϒ cascades, thus impacting [Cl-]i and GABAergic transmission [30].